# Quantum Machine Learning for Classification of Left Ventricular Ejection Fraction Phenotypes from Echocardiograms

Pierre Decoodt
*CHU Brugmann*
*Université Libre de Bruxelles*
Brussels, Belgium
ORCID 0000-0003-4223-6532

Muhammad Waqas Arshad
*Computer Science and Engineering*
*University of Bologna*
Bologna, Italy
ORCID 0000-0003-3262-5833

Marielle Morissens
*Hôpital Universitaire de Bruxelles*
*Université Libre de Bruxelle*
Brussels, Belgium
ORCID 0000-0003-1487-6583

David Q. Liu
*Rosen Center for Advanced Computing*
*Purdue University*
West Lafayette, IN 47907, USA
ORCID 0000-0001-6159-0524

*Abstract*—End-to-end video classification by transfer learning allows one to categorize left ventricular ejection fraction (LVEF) into reduced EF (rEF), midrange EF (mEF), and preserved EF (pEF) from echocardiographic recordings, avoiding delineation of the LV cavity by a human expert or an AI algorithm. We developed a PyTorch implementation using MoViNet. Classical and PennyLane-based classical-quantum models were created. Fine-tuning involved the top four of the five model blocks. We tested our models on Stanford's EchoNet dataset with the original train-val-test split. We tuned the models on the validation set. We developed ternary classifiers for distinguishing between rEF, mEF and pEF, and binary classifiers for rEF vs. rest and not pEF vs. rest. We used the output probabilities of all the classifiers as features subjected to a soft-voting ensemble algorithm consisting of random forest, Gaussian naive Bayes, and logistic regression. For the test set, the extension of the receiver operating characteristic (ROC) to one-vs.-rest multiclass showed a micro-averaged ROC AUC score of 0.96. The ROC AUC score and the balanced accuracy were 0.96 and 0.89 for rEF vs. rest, 0.94 and 0.86 for not pEF vs. rest. After optimization of the decision threshold, the sensitivity and specificity of rEF vs. rest and not pEF vs. rest were always above 0.85.

*Index Terms*—quantum computing, machine learning, ensemble learning, medical imaging, video recording, cardiology

## I. INTRODUCTION

Since an advantage of quantum neural networks over classical CNN was first demonstrated [1], research is ongoing to assess their usability in the interpretation of medical images. In a hybrid neural network, images can be encoded in an initial quantum layer, as in quantum-classical models for the diagnosis of COVID-19 [2] or knee osteoarthritis [3]. Alternately, the quantum layer is inserted in the last part of the model [4]. This hybrid classical-quantum (CQ) approach is proposed for different imaging techniques, such as brain CT for Alzheimer's disease [5], magnetic resonance imaging for the

classification of brain disorders [6], coronary artery disease angiography [7], or chest radiographs for the detection of cardiomegaly [8]. The left ventricular ejection fraction (LVEF) is the percentage of blood present in the left ventricle at the end of diastole that is ejected in systole. LVEF is the parameter most evaluated to assess the functional state of the heart in patients, and echocardiography is the most frequently used imaging technique for this purpose. For diagnosis, treatment, prognosis, and establishment of a path of heart failure care, a ternary ordinal classification of LVEF is recommended, which distinguishes between reduced (rEF), midrange (mEF), and preserved (pEF) LVEF [9].

Numerous machine learning (ML) algorithms have been developed for LVEF evaluation [10]–[12] and several of them were tested on public datasets with a careful human label, such as EchoNet-Dynamic [13] and CAMUS [14]. However, they generally require an initial human and/or ML intervention. This need for on-site human expertise or reliable ML segmentation can be alleviated by an end-to-end approach based on transfer learning, as demonstrated in a pilot study using AutoML in Google Vertex AI [15].

We aimed to develop on open-source software an efficient end-to-end model using transfer learning for the classification of LVEF from echocardiographic video sequences and to compare classical and CQ versions, all based on the same backbone and subjected to the same fine-tuning protocol. To predict the LVEF phenotype, a ternary classifier appears to be the logical option. However, binary straightforward classification of rEF vs. rest or not pEF vs. rest can be considered in a first step, allowing, as a by-product, the ternary classification. We wanted to compare both approaches. Moreover, disposing of six classifiers, classical and CQ, of ternary and binary type, allowed to verify if performance can be improved by ensemble

learning.

## II. Material and methods

A high-level flow chart is presented in Fig. 1.

We used the EchoNet Dynamic database, comprising 10,030 pre-processed, deidentified, and converted echocardiographic video sequences from DICOM to AVI format [13]. We respected the original train/val/test split. To maintain comparability, we did not correct the test set for the label errors identified by a panel of experts in the seminal study. LVEF is usually reported as a value rounded to the nearest integer in clinical reports. Consequently, from the LVEF values obtained by the Simpson method, we defined the true label of each video as follows: rEF for LVEF $< 39.5\%$, pEF for LVEF $\geq 49.5\%$, and mEF otherwise (Table I). Because the models were pre-trained on clips of 50 frames, the AVI videos were normalized to 50-frame MP4 sequences. Data transformation during training and prediction consisted of resizing to $(172 \times 172)$, converting to grayscale, casting to 32-bit floats, and normalization using mean values [0.43216, 0.394666, 0.37645] and standard deviations [0.22803, 0.22145, 0.216989]. These normalization values were those used in MoViNetA0 during pre-training on the Kinetics-600 dataset. This is common practice in transfer learning for medical applications where models pre-trained on colored images work for grayscale ones, such as mammography, MRI, CT scan, or chest X-rays.

We evaluated three types of classifications and used majority downsampling to balance the training set, while maintaining the original unbalanced validation and test sets unchanged (Table II). For the two binary classifiers, the balance is, respectively, for rEF vs. rest and not pEF vs. rest. A further balancing process implied the mEF and the other rest class. This allows sufficient mEF cases for training., which proved advantageous in preliminary test, despite a possible effect on generalization.

The model backbone was MoViNetA0 [16] in its version without stream buffer, pre-trained on the Kinetics-600 dataset. In a first exploration, we tried the models proposed by Pytorch. Many were beyond reach due to lack of sufficient working

### TABLE I
### LVEF Classes in Dataset

| Set | rEF | mEF | pEF |
|---|---|---|---|
| Training | 927 | 673 | 5865 |
| Validation | 151 | 125 | 1012 |
| Test | 151 | 124 | 1002 |
| Total | 1229 | 922 | 7879 |

### TABLE II
### Class Sizes in Training Set After Downsampling

| Model | rEF | mEF | pEF |
|---|---|---|---|
| Binary, rEF vs. rest | 927 | 106 | 821 |
| Binary, not pEF vs. rest | 927 | 673 | 1600 |
| Ternary | 673 | 673 | 673 |

memory. However, the R3D 18, R2Plus1D 18 and Swin3D T were tested in the classical and hybrid versions. They produced poor results, with an ROC AUC below 0.90, even after fine-tuning. MoViNetA0, which also outperformed MoViNetA1, was chosen. The top layers were modified. For the classical model, the output dimension of the last Conv3d layer was changed from 600 to 1 for binary classifiers or 3 for ternary classifiers. For hybrid models (Fig. 1a), we used PennyLane [17]. We used a parameterized quantum circuit (PQC) commonly used in hybrid machine learning applications for medical imaging [5]–[8]. The details are given in Fig. 2. The output dimension of the preprocessing linear layer of the dressed quantum circuit was equal to the number of qubits. A classical activation function (torch.tanh) and a constant np.pi/2.0 scaling followed. For the PCQ, we tested configurations with 4, 6, 8, 10 and 16 qubits and with 4 and 6 repetitions. The best performing version, 10-qubit with 6 repetitions, was chosen. For the initialization of the PQC parameters, we used a random normal distribution of quantum weights with a spread of 0.01. A post-processing linear CNN with input size 10 and output size 1 for binary classifiers or 3 for ternary classifiers completed the dressed circuit. For both classical and hybrid models, the last layer output was submitted to a sigmoid function if binary, or to a softmax function if ternary.

For training and prediction, we used a 16GB NVIDIA Tesla T4 GPU in Google Colab. The training consisted of a "frozen" phase of two epochs, during which only the few last layers had their gradient activated. This was followed by a 12-epoch fine-tuning phase, during which the gradients were activated from the start of block b1 to the last layer. BCEWithLogitsLoss was the criterion for binary classifiers. CrossEntropyLoss was the criterion for ternary classifiers. The optimizer was Adam with an initial learning rate of 0.001. The scheduler was ReduceLROnPlateau with patience set to 0.

Then followed a validation phase using the weights of the epochs for model prediction. The epoch with the highest ROC AUC score in the validation set was identified. For ternary classifiers, the microaveraged ROC AUC score was preferred to the macroaveraged score due to the imbalance of the dataset. The best-epoch weights were used for the prediction of the test set and the assessment of the model performance.

Before comparing the performance of the binary classifiers, we tuned the decision thresholds with the constraint that their value corresponds to the lower positive difference between sensitivity and specificity in the validation set. These thresholds were then applied to calculate the sensitivity and specificity in the test set.

Our ensemble learning approach is summarized in Fig. 1b. For each of the six vanilla classifiers, we first used the best-epoch weights to obtain, in the form of probabilities, the output of the complete dataset. This was followed by balancing the training set of 7865 values. In preliminary tests, oversampling these probabilities was clearly more effective than downsampling. In fact, downsampling for the minority class, mEF, greatly reduced the datasets. Imbalanced-learn ADASYN was chosen among the available algorithms for oversampling because it

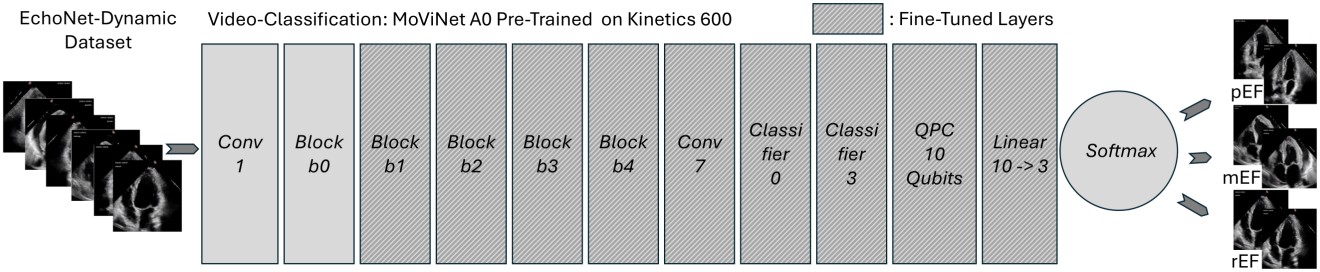

(a) Hybrid CQ Ternary Classifier

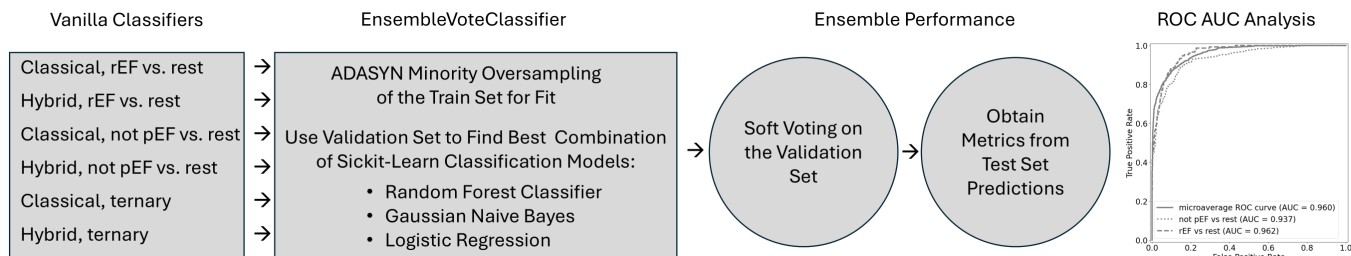

(b) Ensemble Learning

Fig. 1. High-Level Flowchart; (a) an example of classifier based on MoViNetA0; (b): output of six vanilla classifiers submitted to a soft voting algorithm.

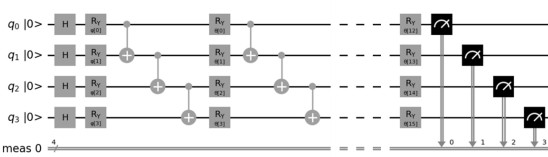

Fig. 2. In this example, a smaller parametrized quantum circuit with four qubits and four variational layers is depicted for better comprehension. First, local embedding consists of superposing each qubit (layer of Hadamard gates "H") and rotating it by an angle $\phi$ in function of the classical input value (first layer of $R_y$ rotations). Then we have a trainable sequence of four variational layers where the qubits undergo full entangling and then $R_y$ rotations by $\theta$ angles. Last, a measurement layer in the Z basis allows for classical output.

exhibited the best performance on the validation set. The size of the training set was increased by oversampling the rEF and mEF classes. The validation and test sets had their size of 1288 and 1277 unchanged. The probabilities were used as features in a meta-classifier for a soft vote ensemble (MLxtend EnsembleVoteClassifier [18]). We tested several sets of vanilla classifiers as entries: the subsets of the four binary, the two ternary, the three classical, the three hybrid, and the complete set of six classifiers.

The scikit-learn classification models considered for incorporation in the EnsembleVoteClassifier were: RandomForest-Classifier, GaussianNB, LogisticRegression (solver: lbfgs), SGDClassifier(loss: modified Huber), KNeighborsClassifier, and DecisionTree. The EnsembleVoteClassifier was fit on the training set. The voting was based on the probabilities observed in the validation set and the final performance estimated from the test set.

We analyzed the response of our not pEF vs. rest ensemble classifier in face of the 40 videos of the Stanford study that were re-evaluated in a blinded review by an five-expert panel. The panel rated that 38% of these videos had significant problems with video quality and acquisition and 13% had significant arrhythmias, making them a good sample to evaluate the robustness of our approach in difficult cases. In this step, we used a 0.5 threshold for our prediction to allow a fair comparison with EchoNet results.

III. RESULTS

In our preliminary heuristic search for the best hyperparameters, we looked at which portion of the model fine-tuning performed better by looking at the monitored metrics. From this observation, an activation of the layers from the start of the MoViNet A0 block 1, including 215 out of 235 layers, appeared optimal in terms of loss, accuracy and ROC AUC score. Fig. 3 shows an example of the metrics values vs. the number of activated layers. Limiting fine-tuning to the last few layers or including the three earliest layers were both counterproductive options.

Table III reports for the six vanilla classifiers the selected epoch with the corresponding ROC AUC score and balanced accuracy for the validation and test sets. Fig. 4 shows an example of score curves observed during the training and validation phases.

The best microaverage ROC AUC scores were obtained by the voting algorithm based on a weighted combination of RandomForestClassifier, GaussianNB, and LogisticRegression. We tested the different possible combinations of vanilla classifiers. The results in different subsets of classifiers (all binary, all ternary, all classical, all hybrids) are presented in Table IV.

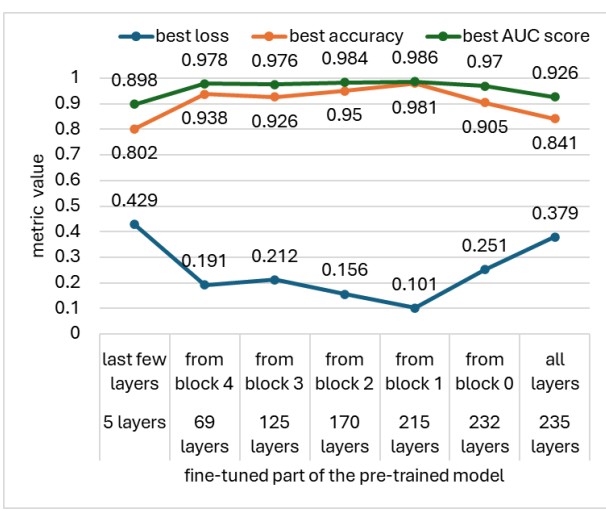

Fig. 3. Monitored metrics during training for the hybrid 10-qubit binary classifier of not pEF vs. rest.

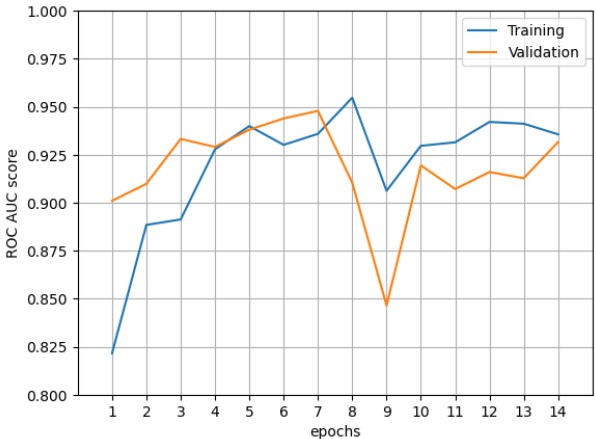

Fig. 4. Evolution of training and validation ROC AUC scores for the hybrid 10-qubit binary classifier of rEF vs. rest. In this example, Epoch 7 was chosen for the model prediction in the test set.

The combination of the six classifiers had the highest scores for the validation and test sets.

Fig. 5 shows the extension of ROC to the one-vs.-rest multiclass, allowing for the evaluation of the performance of the ensemble ternary classifier.

For rEF detection, the advantage of the ensemble ternary classifier over the classical and hybrid versions appears clearly when examining the confusion matrices (Fig. 6). The classical model (Fig. 6a) misclassified more than half of the true rEF cases. The hybrid model (Fig. 6b) is globally better at this task but classifies more cases of true rEF as pEF. The ensemble model (Fig. 6c) is the best for predicting rEF, with 6% of the cases of rEF misclassified as pEF.

Fig. 7 shows cases of correct predictions of pEF and rEF vs. extremely incorrect predictions (rEF instead of pEF and pEF

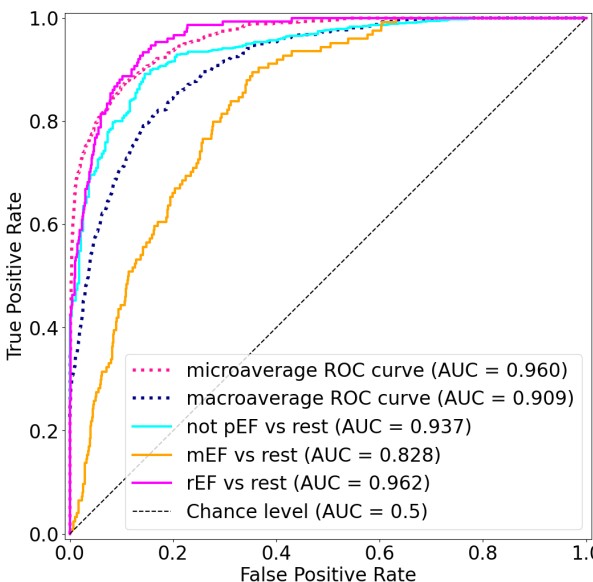

Fig. 5. Extension of ROC to one-vs-rest multiclass applied to the ensemble ternary classifier.

instead of rEF). These examples were drawn at random in each of these four combinations. The great variation in image quality is illustrated. Looking at the corresponding videos confirmed a good imaging quality for Fig. 7a and Fig. 7b, and a poor quality for Fig. 7c and Fig. 7d. It turned out that the video of the incorrect classification of rEF as pEF is among those with the greatest discordance between the prediction of the model and the initial human label of LVEF in the seminal study: the five blinded experts reported poor image quality and unanimously preferred the 35.0% human estimation of LVEF over the 49.0% by the EchoNet dynamic model.

The constrained thresholds that we needed to calculate from the validation set before comparing the ensemble and the vanilla

TABLE III
ROC AUC Score (AUC) and Balanced Accuracy (BA) in Classical (C) and Hybrid (H) Classifiers at Selected Epoch (EP)

| Classifiers | EP | Val AUC | Val BA | Test AUC | Test BA |
|---|---|---|---|---|---|
| C rEF vs. rest | 5 | 0.969 | 0.894 | 0.952 | 0.887 |
| H rEF vs. rest | 7 | 0.948 | 0.866 | 0.925 | 0.855 |
| C not pEF vs. rest | 7 | 0.937 | 0.844 | 0.936 | 0.837 |
| H not pEF vs. rest | 7 | 0.942 | 0.861 | 0.932 | 0.856 |
| C ternary | 10 | 0.956 | 0.833 | 0.947 | 0.814 |
| H ternary | 9 | 0.932 | 0.820 | 0.914 | 0.789 |

TABLE IV
Microaveraged ROC AUC Score

| Sets of classifiers | Validation set | Test set |
|---|---|---|
| all binary | 0.961 | 0.956 |
| all ternary | 0.943 | 0.931 |
| all classical | 0.964 | 0.959 |
| all hybrid | 0.951 | 0.934 |
| all six classifiers | 0.967 | 0.960 |

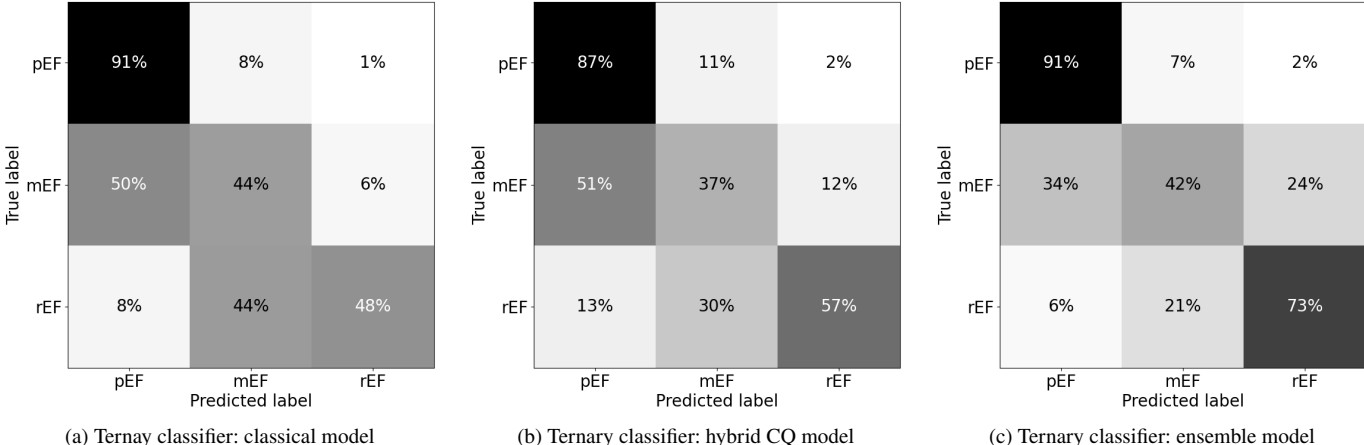

(a) Ternay classifier: classical model     (b) Ternary classifier: hybrid CQ model     (c) Ternary classifier: ensemble model

Fig. 6. Confusion matrices from ternary classifiers. The matrices are normalized over the true labels .

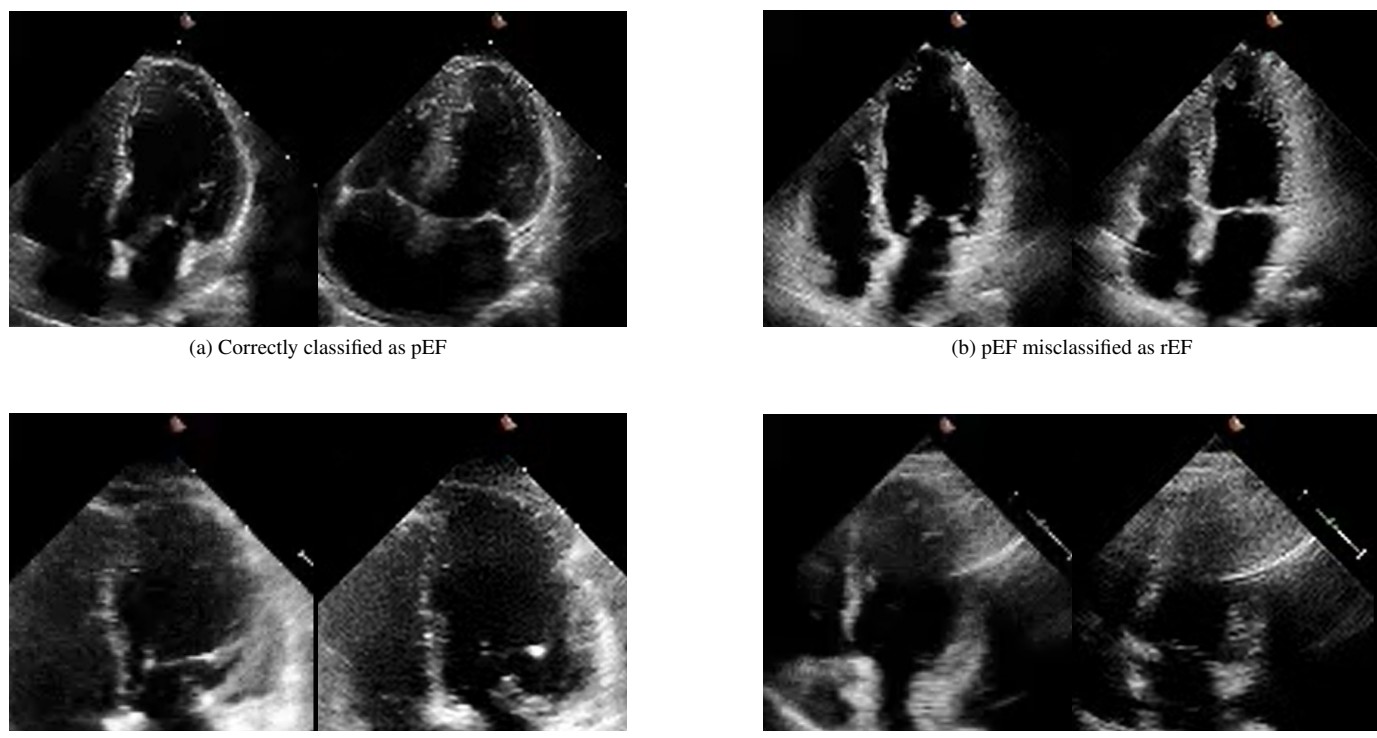

(a) Correctly classified as pEF            (b) pEF misclassified as rEF

(c) Correctly classified as rEF            (d) rEF misclassified as pEF

Fig. 7. End-diastolic and end-systolic frames in correctly and incorrectly classified videos. AVI files and LVEF by Simpson method: (a) 0X4F89846030713617, LVEF 50.8%; (b) 0X59BB0313F22D6C09, LVEF 52.3%; (c) 0X662EF84E75806C27, LVEF 32.1%; (d) 0X1EDA0F3F33F97A9D, LVEF 35.0%.

versions of the binary classifiers are reported in Table V. The values were lower for the ensemble versions.

The comparison of the performance of the ensemble vs. vanilla classifiers for rEF is presented in Table VI. All metrics appear better with the final ensemble classifier.

The comparison of the performance of the ensemble vs. vanilla classifiers for not pEF is presented in Table VII. The metrics are the best for the final ensemble version except for PR AUC, which is the best for the classical version.

Table VIII shows the predictions for the 40 videos that had

TABLE V
DECISION THRESHOLDS FOR BINARY PREDICTION

| Binary classifier | Threshold |
|---|---|
| classical, rEF vs. rest | 0.774 |
| hybrid, rEF vs. rest | 0.883 |
| ensemble, rEF vs. rest | 0.102 |
| classical, not pEF vs. rest | 0.737 |
| hybrid not pEF vs. rest | 0.504 |
| ensemble, not pEF vs. rest | 0.127 |

| Metrics | Classical | Hybrid | Ensemble |
|---|---|---|---|
| Accuracy | 0.883 | 0.868 | 0.904 |
| Balanced accuracy | 0.879 | 0.854 | 0.891 |
| ROC AUC | 0.952 | 0.925 | 0.962 |
| PR AUC | 0.767 | 0.560 | 0.814 |
| Sensitivity | 0.874 | 0.834 | 0.874 |
| Specificity | 0.885 | 0.873 | 0.908 |

TABLE VII
BINARY CLASSIFIER NOT pEF vs. REST

| Metrics | Classical | Hybrid | Ensemble |
|---|---|---|---|
| Accuracy | 0.855 | 0.850 | 0.859 |
| Balanced accuracy | 0.860 | 0.857 | 0.864 |
| ROC AUC | 0.936 | 0.932 | 0.937 |
| PR AUC | 0.836 | 0.821 | 0.823 |
| Sensitivity | 0.869 | 0.869 | 0.873 |
| Specificity | 0.851 | 0.844 | 0.855 |

been re-evaluated by experts in the Stanford study. Taking into account the judgment made by the panel as the truth, a correct prediction was observed in 23/40 videos with the human Simpson method, 28/40 with the EchoNet model and 24/40 with our ensemble classifier. An agreement was observed between the Simpson method and the ensemble model in 29/40 videos. EchoNet and the ensemble models gave the same prediction in 34/40 videos. In the subset of 23 videos where the human label was judged correct, the predictions were similar by EchoNet and the ensemble model in all but one case, incorrectly classified pEF by EchoNet. In the subset with an incorrect human label, EchoNet and the ensemble model diverged in 5/17 instances, with 3 pEF and 2 not pEF videos incorrectly classified by Simpson rule and by the ensemble model.

TABLE VIII
PREDICTIONS FOR DIFFICULT CASES

| Panel | Human | EchoNet | Ensemble | Count |
|---|---|---|---|---|
| not pEF | not pEF | not pEF | not pEF | 13 |
| not pEF | pEF | pEF | pEF | 5 |
| pEF | pEF | pEF | pEF | 4 |
| not pEF | pEF | not pEF | not pEF | 4 |
| pEF | pEF | not pEF | not pEF | 3 |
| pEF | not pEF | pEF | not pEF | 3 |
| pEF | not pEF | pEF | pEF | 2 |
| not pEF | not pEF | pEF | pEF | 2 |
| not pEF | pEF | not pEF | pEF | 2 |
| pEF | not pEF | not pEF | not pEF | 1 |
| not pEF | not pEF | pEF | not pEF | 1 |

## IV. DISCUSSION AND CONCLUSION

Our research was not designed to achieve an AUC score of prediction of not pEF as high as in the Stanford study, where the algorithm required expert human delineation of the left ventricle. In that previous seminal study, no rEF vs. rest or ternary classification was performed.

We built a pipeline where six classifiers were joined to produce a final ternary classifier, which itself allows better binary classification. For clinicians, this can be useful: they can base their judgment on the output probabilities for the three classes, or they may prefer to rely on a prediction of rEF vs. rest or not pEF vs. rest. In fact, in medical applications, the tuning of the decision threshold depends on the clinical context, e.g. detection in a general population, selection of suspected or confirmed cardiac patients, choice of a specific path of health care, and follow-up. Here, we constrained the specificity and sensitivity of a binary classifier to be as similar to each other as possible. This allowed for comparison of the vanilla and ensemble classifiers. Ensemble learning carries the best results, with values that all exceed 0.85 for sensitivity and specificity.

To our knowledge, no hybrid CQ transfer learning has yet been described for end-to-end LVEF classification of echocardiographic video recording. We demonstrate here that this approach works in an application in which the time dimension is added to the imagery. Hybrid CQ models proved to be effective. In a perspective of "beating" classical machine learning, the slightly lower performance observed in these models casts some doubt about their benefit when used alone. However, taking quantum advantage as the right goal for QML is subject to criticism, and alternative research agendas are proposed [19]. The hybrid classifiers differ by adding quantum expressivity, which may explain that the contribution of QML to our ensemble approach is not negligible, as shown in Table VI and Table VII. Our type of doctored quantum circuit was chosen because it is commonly encountered in hybrid transfer learning for medical imaging, but more optimization attempts can be considered. Expressibility and trainability are indeed dependent on several factors, such as the number of parameters and their initialization, the depth of the circuit, and the degree of entanglement [20]–[23]. However, verifying the impact of the many possible PQC modifications through a sound statistical analysis based on k-fold cross-validation would require significant calculation time.

Interpretability analyzes are a possible addition, and showing selected video samples can enhance trustworthiness. The published reports on Grad-Cam equivalents for videos are scarce. A program tested on tennis playing, Video TCAV, is described [24], but uses the Swin Transformer and requires a YOLO-v7 object detector. We did not develop a program adapted to MoViNet. However, this challenging task can be considered in further studies with if possible an evaluation of the video overlead heatmaps by a blind expert panel.

Other refinements can be considered for future research. Generalization properties could be assessed by testing external datasets. Leaving our rigorous protocol based on the original Stanford train-var-test split, a K-fold cross-validation would allow a statistical comparison of classifier performance [25]. There is a possibility that the present end-to-end concept will prove useful in larger-scale projects. For instance, when exploiting more detailed images (DICOM typical resolution is 768*576), or in two-dimensional and three-dimensional imagery, the ML process could be accelerated or simply made possible by using quantum hardware.

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
