# OpenReview forum: "Quantum Machine Learning for Classification of Left Ventricular Ejection Fraction Phenotypes from Echocardiograms"
_IEEE.org/EMBS/BHI/2025/Conference — BHI 2025_

### Official Review · Reviewer_Vu9j · 2025-07-04
**Quantum Machine Learning for Classification of Left Ventricular Ejection Fraction Phenotypes from Echocardiograms**

**Confidence:** 3
**Clarity Of Writing:** good
**Clinical Significance:** fair
**Methodological Novelty:** fair
**Overall Rating:** 6
**Final Rating:** 7

**Experiments And Results:**

good

**Questions For The Authors:**

What is the specific motivation for incorporating quantum machine learning (QML) in this context?

Can you explain how ADASYN-generated synthetic samples were validated for clinical plausibility?

Could you elaborate on the choice and source of normalization values?

**Strengths:**

Methodological Rigor, the paper demonstrates a thorough experimental setup with well-documented training protocols, model architecture details, and systematic evaluation using multiple metrics.

Avoiding the need for manual or AI-based segmentation of the left ventricle, opting for an end-to-end video classification approach that streamlines the diagnostic process and may reduce dependency on expert input.

The inclusion of both classical and hybrid classical-quantum (CQ) models allows for a comparative study, which is a growing area of interest in machine learning and medical imaging.

The combination of output probabilities from multiple models into a soft-voting ensemble improved performance and showed a thoughtful attempt.

The study focuses on a high-impact clinical problem (LVEF classification), and the output formats (e.g., rEF vs. rest) align with real-world decision-making scenarios in cardiology.

Detailed tables and figures (e.g., confusion matrices, ROC curves) support transparency in model performance evaluation.

**Summary Of The Paper:**

The paper presents a framework for classifying left ventricular ejection fraction types-rEF, mEF, and pEF from echocardiographic video sequences using both classical and hybrid classical-quantum machine learning models. The authors utilize the publicly available EchoNet Dynamic dataset and employ the MoViNet-A0 video classification architecture as the backbone for all models. They fine-tune the top layers of the network and apply both binary and ternary classification approaches. Six classifiers are developed: three classical (binary and ternary) and three hybrid CQ versions, all subjected to the same training and evaluation protocols. The output probabilities from these models are further used as features in a soft-voting ensemble meta-classifier composed of classical algorithms. The models are evaluated on the original train/validation/test split using performance metrics such as ROC AUC, etc.

**Weaknesses:**

The motivation for using quantum machine learning is not clearly justified

Key technical concepts, such as the quantum circuit architecture or PennyLane framework, are not explained well, making the paper difficult to follow for readers unfamiliar with quantum computing.

The dataset remains imbalanced even after downsampling, which may affect model generalization, additional analysis or justification is needed.

The use of synthetic oversampling (ADASYN) in a clinical setting raises concerns about the realism and clinical validity of the generated samples.

Relavent literature regarding LVEF classification using ML/DL/QML not cited or discussed

---

### Official Review · Reviewer_z4xm · 2025-07-12
**Quantum Machine Learning for Classification of Left Ventricular Ejection Fraction Phenotypes from Echocardiograms**

**Confidence:** 3
**Clarity Of Writing:** good
**Clinical Significance:** fair
**Methodological Novelty:** fair
**Overall Rating:** 5

**Experiments And Results:**

good

**Questions For The Authors:**

What tangible benefits do the hybrid CQ models offer over classical models, given their slightly lower performance? Could the quantum circuits be optimized further? A clear answer with experiments would raise confidence in the claimed methodological novelty.

Why was MoViNet chosen over other architectures (e.g., R2Plus1D, SlowFast)? Have you benchmarked your classical models against these? Strong comparative results would improve the assessment of the pipeline’s competitiveness.

How does the system handle poor-quality echocardiograms? The paper notes video quality issues (Fig. 6), but does not quantify their impact. Clarifying this could strengthen clinical significance.

Do you plan to test this approach on datasets beyond EchoNet? External validation would greatly increase confidence in generalizability and clinical utility.

**Strengths:**

This paper explores an ambitious and timely topic at the intersection of quantum computing and medical imaging. By proposing an end-to-end pipeline for classifying left ventricular ejection fraction (LVEF) directly from echocardiographic video sequences, it addresses a clinically important challenge, bypassing the need for manual or automated left ventricular segmentation. This aligns well with the trend in recent works, such as Ouyang et al. (Nature, 2020), that advocate for fully automated approaches in cardiac imaging.

One of the notable aspects of the study is its attempt to bring quantum machine learning into the clinical domain. While quantum models are still in their infancy for practical applications, the integration of hybrid classical-quantum (CQ) architectures within a deep learning framework reflects forward-thinking. This effort demonstrates how such models could eventually complement classical methods in scenarios where quantum advantages, like higher expressivity, become realizable on larger datasets or with future quantum hardware.

The authors also take a thoughtful step in applying ensemble learning to combine multiple models, both classical and quantum, to create a robust final classifier. This design choice proves effective, leading to improved performance metrics, particularly for detecting reduced ejection fraction (rEF). Furthermore, the experimental workflow is clearly structured, covering both binary and ternary classification tasks and incorporating decision threshold optimization, a critical detail for potential real-world deployment in cardiology.

**Summary Of The Paper:**

The paper proposes an end-to-end video classification framework to categorize left ventricular ejection fraction (LVEF) into reduced (rEF), midrange (mEF), and preserved (pEF) directly from echocardiographic recordings. The novelty lies in integrating quantum machine learning into this workflow. The authors use MoViNetA0 as the backbone and implement both classical and hybrid classical-quantum (CQ) models using PennyLane. They train binary classifiers (rEF vs. rest; not pEF vs. rest) and ternary classifiers (rEF, mEF, pEF) and combine outputs via ensemble learning (soft-voting with Random Forest, Naive Bayes, Logistic Regression). Experiments are performed on the EchoNet-Dynamic dataset (N=10,030), adhering to the original train/validation/test split. The best ensemble models achieve microaveraged ROC AUC scores of 0.96 and balanced accuracy above 0.89 for binary tasks.

**Weaknesses:**

1 Methodological limitations relative to state of the art

Marginal performance gains from quantum layers: The hybrid CQ models underperform their classical counterparts and contribute little to the final ensemble (e.g., ROC AUC for hybrid ternary = 0.932 vs. classical = 0.956). Prior studies (e.g., Mari et al., Quantum, 2020) highlight CQ models’ potential only on small datasets where quantum expressivity is advantageous. Here, with >10k samples, classical models dominate.

Shallow quantum circuits: Using 10-qubit parameterized quantum circuits is not justified or optimized. Other works (e.g., Schuld et al., PRX Quantum, 2021) explore deeper circuits or variational quantum layers with proven benefits.

No ablation study: There is no investigation into whether the quantum layer, ensemble learning, or MoViNet backbone individually contributes meaningfully to performance improvements.


2 Dataset and experimental design concerns

No external validation: Results are limited to EchoNet-Dynamic, a relatively clean and preprocessed dataset. The lack of testing on independent datasets (e.g., CAMUS, HMC-QU) leaves generalizability unproven.

Class imbalance inadequately addressed: While ADASYN oversampling is used for ensemble training, the impact of imbalanced class distributions on model bias is not thoroughly analyzed.


3 Missing comparisons and context

Insufficient benchmarking: The study does not compare its results to prior EchoNet-based work (e.g., Ouyang et al., Nature, 2020) or recent end-to-end models like R2Plus1D and SlowFast networks, which could contextualize performance gains.

Limited discussion of clinical relevance: Beyond accuracy metrics, there is little discussion of how this pipeline could be used in real-world cardiology workflows, where explainability and trust are crucial.


4 Suggested additional experiments

Ablation experiments - To isolate the impact of quantum layers and ensemble learning.

Cross-validation or K-fold evaluation - To assess model stability and avoid overfitting to the EchoNet split.

Saliency mapping (e.g., Grad-CAM) - To verify that the models focus on the LV region, increasing clinical interpretability.

External validation - To evaluate robustness to variations in imaging protocols and patient demographics.

In summary:
This paper addresses an emerging intersection of quantum computing and medical imaging, presenting a technically sound and ambitious pipeline for LVEF classification. However, relative to state-of-the-art classical methods, the hybrid CQ approach provides minimal added value and lacks rigorous justification for its inclusion. The study would benefit from deeper quantum architectures, external validation, and stronger benchmarking against established video classifiers. At present, the work represents an interesting proof of concept but falls short of demonstrating clear advantages or readiness for clinical application.

---

### Official Review · Reviewer_NUBo · 2025-07-18
**While the paper is methodologically ambitious and results are strong, the clinical interpretability, statistical robustness, and specific benefits of quantum over classical methods remain underexplored.**

**Confidence:** 4
**Clarity Of Writing:** good
**Clinical Significance:** great
**Methodological Novelty:** great
**Overall Rating:** 7

**Experiments And Results:**

good

**Questions For The Authors:**

What is the specific added value of the quantum circuits? Could you provide results showing performance with and without the quantum component in isolation (not just in ensembles)?

How do you account for or mitigate label noise in the test set, particularly in borderline LVEF cases (e.g., 39.5% vs. 49.5%)? This could significantly affect classification near thresholds.

Did you perform any cross-validation or statistical tests to support the robustness of your findings? If not, would results hold under 5-fold or bootstrap resampling?

Could you provide interpretability analyses (e.g., Grad-CAM, attention maps) for at least a few video samples? This would improve trust and understanding for clinicians.

What are the computational implications of integrating QML in this setting? Is it feasible for real-world clinical inference at scale, given the need for quantum simulators or future hardware?

**Strengths:**

End-to-end LVEF classification: Avoids dependency on segmentation or contouring, making it more practical in low-resource settings.

Novel application of QML: Demonstrates the feasibility of integrating quantum circuits into a deep learning pipeline for video-based medical diagnostics.

Ensemble learning enhancement: Effectively combines outputs from multiple classifiers to boost performance, particularly in the detection of rEF.

Comprehensive evaluation: Includes multiple performance metrics, ROC/PR curves, confusion matrices, and decision threshold tuning.

Open dataset and reproducible architecture: EchoNet-Dynamic is a strong benchmark choice; model architecture and data processing steps are clearly described.

Clinical relevance: Focuses on LVEF classification, a core decision point in heart failure diagnosis and management.

**Summary Of The Paper:**

The paper presents a novel pipeline for classifying left ventricular ejection fraction (LVEF) phenotypes- reduced (rEF), midrange (mEF), and preserved (pEF) - from echocardiographic video sequences using both classical and hybrid classical–quantum machine learning models. Leveraging transfer learning on the MoViNetA0 backbone, the authors create six classifiers: three classical and three hybrid (CQ), spanning binary (rEF vs. rest, not pEF vs. rest) and ternary tasks. The classifiers are trained and evaluated on the EchoNet-Dynamic dataset using a standard train/val/test split. Ensemble learning using soft voting further boosts performance, yielding a micro-averaged ROC AUC of 0.96 on the test set. Performance metrics (balanced accuracy, sensitivity, specificity) for both binary and ternary classifications consistently exceed 0.85 in ensemble models, highlighting the utility of the approach. The study also evaluates model misclassifications, threshold tuning, and the contribution of image quality.

**Weaknesses:**

Limited quantum advantage demonstrated: While QML models are included, their performance is consistently lower than classical models, and their added value appears to come only through ensemble aggregation.

No external validation: All results are limited to EchoNet-Dynamic; generalization across devices, populations, or institutions is not assessed.

Lack of interpretability tools: No saliency or attention visualization is used to understand what aspects of the video drive predictions—particularly important in clinical contexts.

Statistical significance unclear: Results are presented without confidence intervals or statistical testing across models or folds.

Imbalanced dataset and potential label noise: While downsampling and expert label errors are discussed, the robustness of the models to label noise is not quantitatively assessed.

No ablation of ensemble components: More clarity is needed on which base classifiers contribute most to the ensemble’s performance.